# Network Delay and Cache Overflow: A Parameter Estimation Method for Time Window Based Hopping Network

**DOI:** 10.3390/e25010116

**Published:** 2023-01-05

**Authors:** Zhu Fang, Zhengquan Xu

**Affiliations:** 1School of Electronic Information, Wuhan University, Wuhan 430064, China; 2State Key Laboratory of Mapping and Remote Sensing Information Engineering, Wuhan University, Wuhan 430079, China

**Keywords:** network security, hopping network, delayed time windows, time window compensation, time window parameter estimation

## Abstract

A basic understanding of delayed packet loss is key to successfully applying it to multi-node hopping networks. Given the problem of delayed data loss due to network delay in a hop network environment, we review early time windowing approaches, for which most contributions focus on end-to-end hopping networks. However, they do not apply to the general hopping network environment, where data transmission from the sending host to the receiving host usually requires forwarding at multiple intermediate nodes due to network latency and network cache overflow, which may result in delayed packet loss. To overcome this challenge, we propose a delay time window and a method for estimating the delay time window. By examining the network delays of different data tasks, we obtain network delay estimates for these data tasks, use them as estimates of the delay time window, and validate the estimated results to verify that the results satisfy the delay distribution law. In addition, simulation tests and a discussion of the results were conducted to demonstrate how to maximize the reception of delay groupings. The analysis shows that the method is more general and applicable to multi-node hopping networks than existing time windowing methods.

## 1. Introduction

With the advancement of information technology, the wide application of the network promotes the development of the social economy, but it also faces serious security problems. In the face of various malicious network assaults such as Trojans, worms, and viruses, these attacks substantially increase the risks of disclosing user information and resulting in property loss. Traditional network protection approaches such as security vulnerability screening, firewalls, intrusion detection, and so on have been investigated to achieve this goal. Those defensive technologies have raised the degree of network protection and are increasingly becoming the typical network defense configuration.

However, network attack techniques are always improving, and current defensive systems are unable to defend against all new types of network assaults, such as Trojan port hopping, hopping proxy, protocol conversion, distributed denial of service (DDoS), and other unknown attacks. To that end, the idea of moving target defense (MTD) [1,2,3,4] was first proposed in the Astronomy Picture of the Day (APOD) project [5], which was presided over by the USA Ministry of Defense (MoD) department. In that project, they provided a brand-new technical route to deal with new attacks. Unlike past cyber security approaches, MTD is devoted to building a dynamic, heterogeneous, and unpredictable network that may avoid, delay, or stop network assaults by increasing the randomness or decreasing the predictability of the system. Furthermore, MTD technologies attempt to create an unpredictable cyber environment by continuously changing the attack surface and reducing exposure to static targets. Because this technology subverts the traditional defense concept of “fixed and die-hard defense” for specific attacks, it is still effective in defending against some uncertain or unknown attacks and is thus referred to as dynamic or active defense technology, whereas the traditional defense technology is referred to as static or passive defense. According to popular belief, genuinely effective network security must comprise both active and passive defense technology.

Network active defense systems are based on hopping networks, and thus hopping networks have been one of the most active research directions, where some of the primary studies on hopping networks are randomization of network parameters [6,7,8,9], etc. Chang et al. [9] randomized the IP addresses of SDNs to improve the randomness of hopping networks; Luo et al. [10] proposed a random port and address hopping (RPAH) mechanism; Lee et al. [11] proposed a UDP/TCP port-hopping method that allows the server parameters to vary with time and a shared secret key function. To prevent attackers from continuously tracking their targets, Fenske et al. [12] proposed a deployment scheme based on MAC address randomization. Since IPv6 has a larger address space, Dunlop et al. [13] proposed an IPv6-based moving target defense (MT6D) technique. These network parameter randomization techniques break the offensive–defensive balance to some extent, making it more difficult for attackers to attack network targets. However, they sacrifice certain network resources when implementing a hopping network, which impacts the network’s performance.

The influence on network performance includes two main aspects: one is the problem of resource congestion, i.e., in order to carry out network hopping, it is inevitable to consume network resources, thus indirectly causing a reduction in network transmission performance; the other is the so-called network delay packet loss problem, i.e., when each node in the network synchronizes hopping, due to transmission delay, some data transmitted with pre-hopping parameters do not reach the receiving node at the time of hopping, and it also cannot be correctly received by the node after hopping, thus causing transmission packet loss. This not only degrades performance but also interferes with the normal network function when network resources are tight, transmission paths are long, or hops are frequent.

The solution to network delayed packet loss is mainly to increase the data reception time (time window) for a period of time compared to the data sending time in order to balance the network transmission delay. For example, in a port-hopping network, Lee [14] makes the data reception time longer than the data sending time by a period of time and uses the overlapping time of the time window of the two periods before and after to receive delayed data from different ports; to further adapt to the network environment through repeated interactions with the receiver, Kong et al. [15] proposed a sliding time window based on an IPv6 address hopping (AHSTW) model, and Ma et al. [16] proposed a dynamic address tunneling model for IPv6, which they based on the results of the interaction feedback and made the length of the data-reception time window larger than the length of the data-sending time window. These techniques are mainly used in end-to-end networks, and they are tightly coupled to the service, which to some extent severely disrupts normal network operations and is therefore not suitable for general network scenarios.

To address the above problems, we propose a delayed time window estimation algorithm. Firstly, we introduce the concept of delay time window and the estimated value of the delay time window; secondly, we propose the basic conditions for avoiding non-interaction and IP address conflicts and mathematically prove that the time window compensation mechanism can satisfy these conditions, and we give the calculation of the estimated value of the delay time window; that is, after analyzing the network transmission of different data services, we obtain the expression for the estimated value of the delay time window, and at the same time, mathematically, we prove that the estimated value of the delay time window is theoretically close to the optimal value of the delay time window and finally provide the implementation algorithm. The algorithm not only minimizes the loss of delayed data but also does not require interaction with the service. In addition, the method allows the network to consume fewer network resources and is able to maintain normal system service under network resource constraints, and it is therefore applicable to networks in general and is no longer limited to end-to-end networks.

## 2. Preliminaries

### 2.1. Hopping Network

As shown in Figure 1, assume that the network *G* is made up of physical nodes *N*_1_, *N*_2_, …, *N_z_*. The network parameters (IP address, port number, etc.) configured for the corresponding node *N_i_*, *i* = 1, 2, …, *z* is denoted as *h_ij_*, *j* = 1, 2, …, *ξ*. Typically, the set ***H*** = (*h_ij_*)*_z×ξ_* remains constant during the work, while nodes can rely on ***H*** for mutual access. For each node, it is specified that at every period *T*, a parameter is selected from ***H*** and configured to it by some rule, and its network parameters are changed every period *T*. For this change, it is called network hopping, and *T* is the hopping period.

### 2.2. Time Window

As network parameters constantly change in the hopping, a particular network parameter is only valid for a specific period, which is the time window for that network parameter, as shown in Figure 2. Then the network parameter *h_ij_* corresponding to the *i* node at the *j* hopping period is valid for the time range [*tw_a_*(*j*), *tw_b_*(*j*)] for the *h_ij_* time window, denoted as *TW*(*h_ij_*). Since the network nodes are generally synchronous hopping, the time windows of the parameters corresponding to each node are fully overlapping, i.e., *TW*(*h*_1_*_j_*) = *TW*(*h*_2_*_j_*) = … = *TW*(*h_zj_*) = [*tw_a_*(*j*), *tw_b_*(*j*)], so they are denoted uniformly as *TW*(*j*). In the normal case, the time window within which the data is sent is the sending time window *STW*(*j*)= [*st_a_*(*j*), *st_b_*(*j*)]. After receiving the data within the time window, the time window for receiving the data can be denoted as receiving time window *RTW*(*j*) = [*rt_a_*(*j*), *rt_b_*(*j*)]. The set of times at which data arrives after it has been sent within the time window is the arrival-data time window *ATW*(*j*) = [*at_b_*(*j*), *at_b_*(*j*)]. In the ideal case of no network delay, *RTW* (*j*) = *STW*(*j*) = *ATW*(*j*).

### 2.3. Delayed Packet Loss

In hopping networks, some mechanisms are usually designed so that all legitimate nodes can sense the change of ***H*** in real time, so the regular communication of legitimate nodes is unaffected. However, some transmission packet loss is still generated during a short period of time when ***H*** is hopped, which is called delayed packet loss. As shown in Figure 3, the specific cause of delayed packet loss is that *ATW*(*j*) is slightly delayed by a period of time compared to *STW*(*j*) due to the presence of network delay *d*, which results in *RTW*(*j*) not being able to cover *ATW*(*j*) completely. As a result, different methods have been tried to make *RTW*(*j*) cover *ATW*(*j*)as much as possible. We summarize the descriptions in the literature [15,16,17], where they present two methods for *RTW*(*j*) to cover *ATW*(*j*). One is to maximize *RTW*(*j*) coverage of *ATW*(*j*) by extending *RTW*(*j*) on the basis that *STW*(*j*) does not change [14]; the other is to shorten *STW*(*j*) on the basis that *RTW*(*j*) does not change, which is equivalent to delaying *RTW*(*j*) to give it a chance to cover *ATW*(*j*) completely [15,16], but both methods have drawbacks.

### 2.4. Problem Statement

These two methods have two disadvantages:

One is the problem of conflicting IP addresses. When extending *RTW*(*j*) for some time, then *RTW*(*j*) is a period longer than *STW*(*j*), and there will be a period of overlap between this extra time *ODRT* = [*rt_a_*(*j* + 1), *rt_b_*(*j*)] and *RTW*(*j* + 1). Since the overlapping time can receive data from different ports but not from different IP addresses, receiving data during the overlapping time can lead to a problem of conflicting IP addresses, as shown in Figure 4.

The other is the problem of tight business coupling. When *STW*(*j*) is shortened by a period of time, then a period of time *FDTT* = [*st_b_*(*j*), *st_a_*(*j* + 1)] for which *STW*(*j*) is less than *RTW*(*j*) can be used without being used to send data and without the need to interact with the tasks to obtain *st_b_*(*j*) and *st_a_*(*j* + 1) for this period of time through the time window. The aim is to allow delayed data to arrive in total, and such a time window is mainly used in end-to-end networks (see Figure 5).

## 3. Proposed Scheme

In this section, we present a scheme for a delayed time window compensation mechanism and its parameter estimation to solve the problem of IP address conflicts and tightly coupled services. The implementation of such a mechanism then requires two conditions: firstly, a time window compensation mechanism and defined associated parameters that meet the above requirements; secondly, such a mechanism can effectively perform the function as a time window when the influence of external factors (it is assumed that there are no network attacks on the hopping network and that the hopping network is hopping at a uniform rate) on the compensation mechanism is negligible.

### 3.1. Delay Time Window Compensation Mechanism

According to the above, there are problems with both methods of *RTW*(*j*) covering *ATW*(*j*) by extending *RTW*(*j*) on the basis that *STW*(*j*) does not change length or shortening *STW*(*j*) on the basis that *RTW*(*j*) remains unchanged, so we propose the idea of keeping *STW*(*j*) equal to *RTW*(*j*) and delaying *RTW*(*j*) to *STW*(*j*) for a period of time, with the aim of allowing *RTW*(*j*) to override *ATW*(*j*). This not only avoids IP address conflicts and tight task coupling, but it also solves the problem of delayed packet loss. The proof is as follows:

Firstly, when *STW*(*j*) = *RTW*(*j*), then there is not a period of time when *RTW*(*j*) is longer than *STW*(*j*); this period of time does not create an overlap with *RTW*(*j* + 1), and therefore this does not lead to IP address conflicts. Furthermore, there is also not a period of time when *STW*(*j*) is less than *RTW*(*j*), which indicates that the time window does not require interactive data tasks to obtain such a period of time, and therefore this does not lead to tight service coupling.

Secondly, when *RTW*(*j*) = *ATW*(*j*), then all the data from *STW*(*j*) can be received in *RTW*(*j*), including the delayed data.

Thus, we let *STW*(*j*) = *RTW*(*j*) and *RTW*(*j*) = *ATW*(*j*); they can be achieved by the following two steps:(1)For *RTW*(*j*) = *STW*(*j*), let *rt_b_*(*j*) − *rt_a_*(*j*) = *st_b_*(*j*) − *st_a_*(*j*), then *rt_b_*(*j*) = *st_b_*(*j*) and *rt_a_*(*j*) = *st_a_*(*j*).(2)For *RTW*(*j*) = *ATW*(*j*), assume that *d* is the network delay and *w*^~^ denotes a period of time (delay time window). Under the condition that step (1) and (2) are satisfied, it follows that *at_a_*(*j*) = *st_a_*(*j*) + *d*, *at_b_*(*j*) = *st_b_*(*j*) + *d*, *rt_a_*(*j*) = *st_a_*(*j*) + *w*^~^ and *rt_b_*(*j*) = *st_b_*(*j*) + *w*^~^. Conversely, the values of *rt_a_*(*j*) and *rt_b_*(*j*) can be determined as long as the estimated value of *w*^~^ is obtained and *w*^~^ = *d*, again satisfying *STW*(*j*) = *RTW*(*j*) and *ATW*(*j*) = *RTW*(*j*).

In summary, the delay time window compensation mechanism has the characteristics of necessity and possibility:Necessity means that a delay time window compensation mechanism must solve the problem of delayed packet loss and related problems.Possibility means that there may be a suitable delay time window length, i.e., a value of *w*^~^ (see Figure 6).

### 3.2. Estimation for Delay Time Window

In this section, we obtain the value of *w*^~^ by calculation, as getting the value of *w*^~^ helps us to determine the values of *rt_a_*(*j*) and *rt_b_*(*j*). Since *d* is unknown, then we can only find a way to estimate a value for the network delay and use it as an estimated value for *w*^~^. From this, we present an example with multiple (*l*) data tasks distributed in an observable wholly connected network. By progressively computing the network transmission delay for the typical data tasks in the example, we finally obtain an estimated value of the network delay for *w*^~^. These typical data tasks include an end-to-end data task, a multi-node data task, and multiple multi-node data tasks. This regular data task is more complex, so we plan to start with the simpler data tasks, as shown in Figure 7.

(1) A simple data task. Under normal situations, routers and network agents have sufficient network cache [17,18] to store and forward data, so in most cases, the network cache is still significantly effective in reducing delay in packet loss. However, a few instances can still cause packet loss problems for some data. These are the rare cases where the processor does not have enough processing capability due to large amounts of data or unusual data, and therefore the network cache is insufficient. The few cases that cause delayed packet loss are then our target. Consequently, we examine task 1 (*r* = 1), as shown in Figure 8. In task 1, we start with two nodes, with a network delay of *d*_10_ between node A and node B. When data is sent from A to B, a small amount of data is lost, except for most of the data that is received by B. This is because when B has sufficient network cache and *w*^~^ > *d*_10_, the network cache can store data arriving before *rt_a_*(*j*) without data loss. However, when B does not have sufficient network cache due to oversized data, data arriving before *rt_a_*(*j*) is lost; furthermore, when *w*^~^ < *d*_10_ and *RTW*(*j*) cannot completely cover *ATW*(*j*), the data arriving after *rt_b_*(*j*) are lost (see Figure 9). The lost data can be expressed as:*L*_AB_ = *v*·|*d*_10_ − *w*^~^|(1)
Let *L*_AB_ = 0, *w*^~^ = *d*_10_.

Where *v* denotes the transmission rate, and *L*_AB_ denotes the number of delayed packet losses.

(2) Depicts a multi-node data task. We observe this for task 2 (*r* = 2), where the variable *m* combined with the corresponding subscript number *r* indicates the number *m_r_* of nodes of the corresponding data task, e.g., the number of nodes for the data task 2 (*r* = 2) is *m*_2_. The variable *d* combined with the corresponding subscript numbers *r* and *i* denotes the network delay *d_ri_* of the corresponding task *r* and node *i*, e.g., the network delay for task 2 and node 1 is *d*_21_, as shown in Figure 10. The data tasks for multiple nodes are more complex because data passing through multiple nodes will generate network delays, and there is an accumulation of multiple network delays starting with three nodes. The network transmission of three nodes can be seen as two end-to-end network transmissions, i.e., node 1 to node 2 and node 2 to node 3. As the data causes one network delay after passing node 2 and another network delay when the data reaches node 3, then the sum of the two network delays is the total network delay when the data comes to node 3. We can then calculate the number of packet losses based on the total network delay, but it needs to be clear that Equation (1) already provides the delayed packet loss for the first end-to-end network transmission, so the delayed packet loss for the second end-to-end network transmission is cumulative on top of that (see Figure 11). The delayed packet loss for the three nodes can then be expressed as: *L*_2_ = *v*·|*d*_21_ − *w*^~^|+ *v*·|*d*_21_ + *d*_22_ − *w*^~^|(2)
Based on the delayed packet loss for three nodes, we introduce the delayed packet loss for *m*_2_ nodes, which can be expressed as:*L*_2_ = *v*·|*d*_21_ − *w*^~^|+ *v*·|*d*_21_ + *d*_22_ − *w*^~^|+^…^+ *v*·|*d*_21_ + *d*_22_+^…^+ *d*_2_*_m_*_2_ − *w*^~^|(3)
Combining for (3), we get:$$L_{2}=\sum_{i=1}^{m_{2}}v\cdot \left|\sum_{\kappa =1}^{i}d_{2\kappa }-w^{\sim }\right|$$
Let $d_{2i}=\sum_{\kappa =1}^{i}d_{2\kappa }$, we have:$$L_{2}=\sum_{i=1}^{m_{2}}v\cdot \left|d_{2i}-w^{\sim }\right|$$

Based on Equation (3), we get:(4)$$f_{2}=\frac{L_{2}}{T\cdot v}$$
where *f*_AB_ is the function of packet loss rate for hopping period *T*. 

Let $\frac{\partial f_{2}}{\partial w^{\sim }}=0$, we have:$$\begin{array}{l} \partial \frac{\left(\frac{L_{2}}{v\cdot T}\right)}{\partial w^{\sim }}=0\\ \frac{\partial \left(\frac{{\left(d_{21}-w^{\sim }\right)}^{2}}{T}\right)}{\partial w^{\sim }}+\frac{\partial \left(\frac{{\left(d_{22}+d_{22}-w^{\sim }\right)}^{2}}{T}\right)}{\partial w^{\sim }}+\cdots +\frac{\partial \left(\frac{{\left(d_{21}+d_{22}+\cdots +d_{2m}{}_{{}_{2}}-w^{\sim }\right)}^{2}}{T}\right)}{\partial w^{\sim }}=0\\ w^{\sim }=\frac{1}{m_{2}}\left[m_{2}d_{21}+\left(m_{2}-1\right)d_{22}+\cdots +d_{2m_{2}}\right] \end{array}$$

(3) Depicts multiple data tasks for the whole network. In the observable network, any two or multiple nodes in the network are connected as a single path. Thus, there are multiple paths (*α*) across the network, and in most cases, there are multiple data tasks (*l*) on a single path. Then, after observing the data tasks on this one path, we can further examine the data tasks on multiple paths. Since data tasks on multiple paths are more complex, we first examine data tasks on one path. By observation, accumulating the network delay for each data task can be used as the total network delay for the data tasks on this path. Then similarly, based on (4), accumulating the delayed packet loss for multiple data tasks on the path one can be used as the total delayed packet loss for this path (see Figure 12). The total delayed packet loss on path one can be expressed as:(5)$$\sum_{r=1}^{l}L_{\mathrm{AB}}\left(r\right)$$
Based on the delayed packet loss for path 1, we introduce the delayed packet loss for the α path expressed as:(6)$$L_{l}=\sum_{r=1}^{l}L_{\mathrm{AB}}\left(r\right)$$
Based on (6), the data packet loss rate for period (*T*) can be expressed as:(7)$$f_{l}=\frac{L_{l}}{v\cdot T}$$
Let $\frac{\partial f_{l}}{\partial w^{\sim }}=0$, we have:$$\begin{array}{l} \frac{\partial f_{\alpha }}{\partial w^{\sim }}=\partial \frac{\left(\frac{L_{l}}{v\cdot T}\right)}{\partial w^{\sim }}=0\\ \partial \frac{\left(\sum_{r=1}^{l}\left(\frac{{\left(d_{d_{r1}}-w^{\sim }\right)}^{2}}{T}+\frac{{\left(d_{r1}+d_{r2}-w^{\sim }\right)}^{2}}{T}+\cdots +\frac{{\left(d_{r1}+d_{r2}+\cdots +d_{rm}{}_{{}_{r}}-w^{\sim }\right)}^{2}}{T}\right)\right)}{\partial w^{\sim }}=0\\ \sum_{r=1}^{l}\left[\left(d_{d_{r1}}-w^{\sim }\right)+\left(d_{r1}+d_{r2}-w^{\sim }\right)+\cdots +\left(d_{r1}+d_{r2}+\cdots +d_{rm}{}_{{}_{r}}-w^{\sim }\right)\right]=0\\ \sum_{r=1}^{l}\left[m_{r}d_{d_{r1}}+\left(m_{r}-1\right)d_{r2}+\cdots +d_{rm}{}_{{}_{r}}-m_{r}w^{\sim }\right]=0\\ w^{\sim }=\frac{1}{\sum_{r=1}^{l}m_{r}}\sum_{r=1}^{l}\left[m_{r}d_{r1}+\left(m_{r}-1\right)d_{r2}+\cdots +d_{rm_{r}}\right] \end{array}$$

The estimation of $w^{\sim }$ is to let $w^{\sim }=\frac{1}{\sum_{r=1}^{l}m_{r}}\sum_{r=1}^{l}\left[m_{r}d_{r1}+\left(m_{r}-1\right)d_{r2}+\cdots +d_{rm_{r}}\right]$. When $w^{\sim }=\frac{1}{\sum_{r=1}^{l}m_{r}}\sum_{r=1}^{l}\left[m_{r}d_{r1}+\left(m_{r}-1\right)d_{r2}+\cdots +d_{rm_{r}}\right]$, then $L_{a}=0$ satisfies 0 packet loss for data tasks on *r*. According to [19] as the basis for $w^{\sim }$ estimation, the results obtained by Bolot’s [19] tests are consistent with those obtained by [20,21,22] using simulation and experimental methods. Under the assumption that using bulk traffic for large packets and traffic for small packets in internet traffic estimation is consistent, the structure of the delay time distribution can be described as the relationship between the waiting time *w_n_* and *w_n_*_+1_ for packets *n* and *n* + 1. It influences the network traffic (bits) *b*, the packets (bits) *P*, the service rate of the network (bits/ms) *μ*, and the packet queuing time *δ*. Their relationship can be expressed as $w_{n+1}-w_{n}=\left(b+P\right)/\mu -\delta$. Taking Figure 13 as an example, it shows the distribution of $w_{n+1}-w_{n}-\delta$ for *n* (*n* ≤ 800) UDP (32 bts) packets with $w_{n+1}-w_{n}-\delta$ at *δ* = 20 ms. $w_{n+1}-w_{n}-\delta$ is the network load received by the server within [*nδ*, (*n* + 1)·*δ*] and is measured in ms. From Figure 13, it can be seen that the time is mainly distributed in the area covered by the dashed line. Therefore, our proposed strategy is to select the larger data tasks. This is because they take up more time (delay time = receive time ($at_{r\kappa }$) − send time ($st_{r\kappa }$)) through multi-hop routes [23,24]. For example, in Figure 13, assuming that the largest data task *r* = 2, *i* = 1, …, *m*_2_, then the maximum delay time is $m_{2}d_{21}+\left(m_{2}-1\right)d_{22}+\cdots +d_{2m_{2}}$, and the amount of lost data is $L_{2}=\sum_{i=1}^{m_{2}}v\cdot \left|\sum_{\kappa =1}^{i}d_{2\kappa }-w^{\sim }\right|$ (see Equation (3)). Let $d_{2i}=\sum_{\kappa =1}^{i}d_{2\kappa }$, $d_{2\kappa }=at_{2\kappa }-st_{2\kappa }$. When $w^{\sim }=d_{2i}$, then $L_{2}=0$, i.e., $w^{∼}=\sum_{\kappa =1}^{i}at_{2\kappa }-st_{2\kappa }$*t*. Our approach is to use max{$\sum_{\kappa =1}^{i}at_{2\kappa }-st_{2\kappa }$, 0} as an estimate for *w*^~^, i.e., $w^{\sim }$ = max{$\sum_{\kappa =1}^{i}at_{2\kappa }-st_{2\kappa }$, 0}, $w^{\sim }\subset$ {0, $\left(b+P\right)/\mu -\delta$}.

## 4. Performance

In this section, we give the error and experimental evaluation of our two proposed delay time window schemes. For error evaluation, we mainly analyze the difference between estimated and optimal values of the delay time window. For experimental evaluation, we test the data loss rate of our schemes.

### 4.1. Error Evaluation

In this section, to assess the validity of our proposed estimates of the delay time window, we thus present the error assessment of the delay time window. The error is a key part of the error assessment. The error is defined as *δ* = *w*^~^ − $\bar{w}$, and the mathematical expectation of the error (*E*(*δ*)) reflects the magnitude of the mean of the error between the estimated and optimal values of the delay time window. Therefore, it can be used as an indicator for error assessment. In the error evaluation, let the variables *x*_1_, *x*_2_, …, *x*_2_*_k_*_+1_ denote the network delay, and *x*_1_, *x*_2_, …, *x*_2_*_k_*_+1_ ~ *N* (0,1), and let them be independent of each other, $\bar{w}=\left(x_{1},x_{2},\;\ldots ,x_{2k+1}\right)/2k+1$. Let *med* (*x*_1_, *x*_2_, …, *x*_2_*_k_*_+1_) denote the function that returns an optimal value based on the proposed method. According to [25,26], the probability density distribution for end-to-end delayed packet loss shows gamma distribution. Thus, we consider that its probability density can be viewed as density function $f\left(t\right)=\frac{1}{\sqrt{2\pi }}e^{-\frac{t^{2}}{2}}$, and its distribution may be viewed as $F\left(x\right)=\int_{-\infty }^{x}\frac{1}{\sqrt{2\pi }}e^{-\frac{t^{2}}{2}}dt$. Firstly, the analysis of multiple network delays is more complex, so we start with three network delays, *x*_1_, *x*_2_, and *x*_3_. According to this, we have $\delta =\left(x_{1}+x_{2}+x_{3}\right)/3-med(x_{1},x_{2},x_{3})$.
$$\begin{array}{l} E\left(\delta \right)=\underset{x_{1}<x_{2}<x_{3}}{∭}\left(x_{1}+x_{2}-2x_{2}\right)f\left(x_{1}\right)f\left(x_{2}\right)f\left(x_{3}\right)dx_{1}dx_{2}dx_{3}\\ =\int_{-\infty }^{\infty }f(x_{2})\left[\int_{x_{2}}^{\infty }f\left(x_{3}\right)dx_{3}\int_{-\infty }^{x_{2}}x_{1}f\left(x_{1}\right)dx_{1}+\int_{-\infty }^{x_{2}}f\left(x_{1}\right)dx_{1}\int_{x_{2}}^{\infty }x_{3}f\left(x_{3}\right)dx_{3}-2x_{3}\int_{x_{2}}^{\infty }f\left(x_{3}\right)dx_{3}\int_{-\infty }^{x_{2}}f\left(x_{1}\right)dx_{1}\right]dx_{2}\\ =\int_{-\infty }^{\infty }f\left(x_{2}\right)\left[\left(1-F\left(x_{2}\right)\right)\left(-f\left(x_{2}\right)\right)+F\left(x_{2}\right)f\left(x_{2}\right)-2x_{2}\left(1-F\left(x_{2}\right)\right)F\left(x_{2}\right)\right]dx_{2}\\ =\int_{-\infty }^{\infty }\left[f^{2}\left(x_{2}\right)\left(2F\left(x_{2}\right)-1\right)+2x_{2}\left(F\left(x_{2}\right)-1\right)F\left(x_{2}\right)\right]dx_{2}\\ =\int_{-\infty }^{\infty }\left[f^{2}\left(x_{2}\right)\left(2F\left(x_{2}\right)-1\right)\right]dx+\left(F\left(x_{2}\right)-1\right)F\left(x_{2}\right)f(x_{2}){|}_{-\infty }^{\infty }-\int_{-\infty }^{\infty }f^{2}\left(x_{2}\right)2x_{2}\left(F\left(x_{2}\right)-1\right)dx_{2}\\ =0 \end{array}$$
Similarly, *δ* = *med* (*x*_1_, *x*_2_, …, *x*_2_*_k_*_+1_) − $\left(x_{1},x_{2},\;\ldots ,x_{2k+1}\right)/2k+1$.
$$\begin{array}{l} E\left(\delta \right)=E\left(\left(\sum_{i=1}^{2k+1}x_{i}/2k+1\right)-x^{*}\right)\\ =\int_{x^{*}=med\left(x_{1},x_{2},\ldots ,x_{2k+1}\right)}\left(\sum_{i=1}^{2k+1}x_{i}-\left(2k+1\right)x^{*}\right)f\left(x_{1}\right)f\left(x_{2}\right)\cdots f\left(x_{2m+1}\right)dx_{1}dx_{2}\cdots dx_{2m+1}\\ =\int_{-\infty }^{\infty }{\left\{k{\left[1-F\left(x^{*}\right)\right]}^{k}F^{k-1}\left(x^{*}\right)\left(-f\left(x^{*}\right)\right)+k{\left[1-F\left(x^{*}\right)\right]}^{k-1}F^{k}\left(x^{*}\right)f\left(x^{*}\right)-kx^{*}F^{k}\left(x^{*}\right)\left[1-F\left(x^{*}\right)\right]\right\}}^{k}f\left(x^{*}\right)\\ =\int_{-\infty }^{\infty }k{\left[F\left(x^{*}\right)\left(1-F\left(x^{*}\right)\right)\right]}^{k-1}\left[2F\left(x^{*}\right)-1\right]f^{2}\left(x^{*}\right)dx^{*}-\int_{-\infty }^{\infty }kx^{*}{\left[F\left(x^{*}\right)\left(1-F\left(x^{*}\right)\right)\right]}^{k}f\left(x^{*}\right)dx\\ =0 \end{array}$$
Therefore, *E*(*δ*) = 0, from which it follows that the estimated value of *w*^~^ is an unbiased estimated value.

### 4.2. Experimental Evaluation

The experimental assessment consists of two parts, one for the simulation experiment and the other for the actual examination.

For the simulation experiments, we developed the simulated hopping network program SHN (similar to NS-2) using Dev-C++5.11 and C++, which consisted of seven hopping network nodes, including one transmitter node, one receiver node, and five intermediate nodes, as well as network delays (X) and 14 IP addresses (192.168.1.10 to 192.168.1.13). The host with the SHN is DESKTOP-B4VQAPP, which was configured with CPU Intel (R) Xeon (R) e-2124 3.31ghz, 16GB of RAM, and Windows 10 OS. The experiments were implemented on DESKTOP-B4VQAPP. In our experiments, we first set up the system’s initial values, including the network hopping period *T* = 2000, the delay time window *w* = 50, the number of intermediate nodes *σ* = 5, the mean value *mu* = 50 of the random variable X, and the variance *sigma* = 5, 10, 15, 20, bandwidth (100 Mbit/s). Our first experiment was an end-to-end hopping network packet loss rate experiment. The second experiment was a packet loss rate experiment for a multi-node hopping network. In both experiments, we sent 10^5^ data (32 bytes) from the sending node to the receiving node and counted the data loss rate at period T based on the data received by the receiving node after the data reached the receiving node. All experiments were repeated 5000 times. For end-to-end networks, as shown in Figure 14 and Table 1, because in this method, the size of the time window was set by sending interaction information to the receiving node and feedback from it. The methods of [15,16,17] are better than the method presented in this paper. In addition, as shown in Figure 15, we experimented with the proposed delayed time window approach, and the experimental results show that the packet loss rate is better for time windows of 10 ms, 20 ms, and 30 ms compared to no time window.

For the actual examination environment, the physical connection topology of the network is shown in Figure 16. The system is composed of two hopping subnets; hopping subnet 1 and hopping subnet 2 are connected through the IP bearer network. After the start of the network hopping, the source address and source port of the IP packet of the data platform in hopping subnet 1 after the hopping process are transmitted through the two WAN ports of the S5700 router through the IP bearer network to the hopping equipment in hopping subnet 2 after restoration, and then they are transmitted by the visitors in the data plane, thus completing an ordinary network transmission process.

The test environment is built with two hopping subnets, each of which is connected through an IP bearer network. Each hopping subnet is simulated by the relevant equipment in a physical cabinet. Each cabinet includes eight physical servers and three switches; the eight servers, respectively, achieve service hopping, network hopping, posture display, and other functions. The IP bearer network is simulated by an independent cabinet. The IP bearer network consists of four switches with three-layer routing function. The hardware equipment of the test environment mainly includes servers, switches, routers, etc., whose functions and performance indicators are shown in Table 2.

The maximum network transmission rate is 100 Mbit/s under the national standard for category 5 network cables. Network hopping is initiated at one hop/10 s, one hop/5 s, and one hop/2 s, respectively. In addition, a network transmission stress test was conducted (see Figure 17. Moreover, the performance test of the router hopping was carried out at a network transmission pressure of 500 Mb/s (Figure 18 and Figure 19). The experimental results show that the performance of the router still has a relatively large improvement based on various indicators with the use of delay time windows.

## 5. Summary and Future Research

In this paper, we construct a multi-node delay time window estimation method. The delay time window and the delay time window compensation mechanism are proposed in the method, as well as the estimation of the delay time window length. A series of experiments were performed to test the effectiveness of the delay time window estimation method. In the SHN simulation experiments, the network transmission packet loss rate was less than 0.6% at 50 ms network delay. At 100 ms network delay, the network transmission packet loss rate was less than 1.1%. In the actual test, the test environment network cable was lower than the super category five standard, and the maximum network transmission rate was 100 Mbit/s. Taking the network transmission speed pressure of 500 Mbit/s as an example, the packet loss rate of network transmission using the delay time window method is less than 0.8% under 30 ms network delay. This is an improvement compared to the packet loss rate of network transmission without delay time windows. This proves the effectiveness of the method in this paper.

The first suggestion for future research relates to the later ones concerning the plausibility of the period of change of the network parameters. In the delay time window estimation method, an estimate of the delay time window is proposed based on the calculated value of the network delay for data services. This reduces the network transmission packet loss rate to a certain extent. However, many factors affect the network transmission packet loss rate, including in hopping networks; one of the important factors is the effect of the hopping period on the network transmission packet loss rate. Therefore, the second recommendation for future research is that we will consider setting a reasonable hopping period. There are two main aspects of a reasonable period. The first is that setting the hopping period length should not consume too many system resources or cause too much packet loss. The second is that setting the hopping period length should not have too great an impact on the resistance to external attacks. It is necessary to analyze the effect of the hopping period on these factors and then propose a reasonable hopping period scheme. In future research, we will continue to evaluate the impact of the hopping period on the transmission performance of the system.

## 6. Patents

Zhengquan Xu, Zhu Fang. A method for calculating delay time windows in multi-node networks of hopping networks. National Invention Patent, Patent number 2021107766.

## Figures and Tables

**Figure 1 entropy-25-00116-f001:**
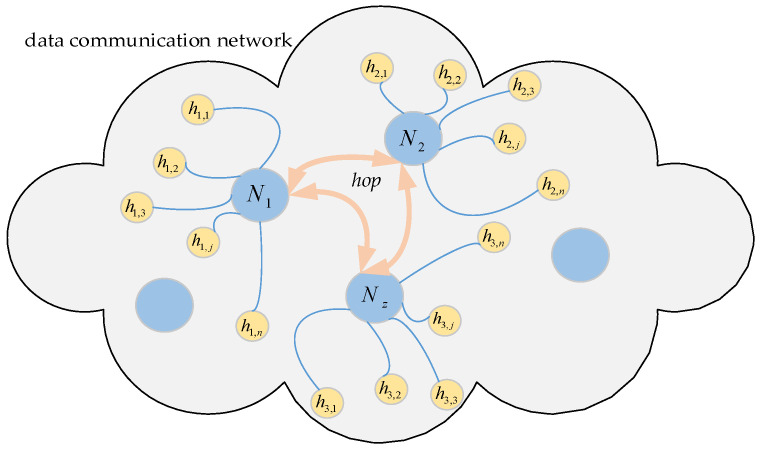
Schematic diagram of the hopping network.

**Figure 2 entropy-25-00116-f002:**
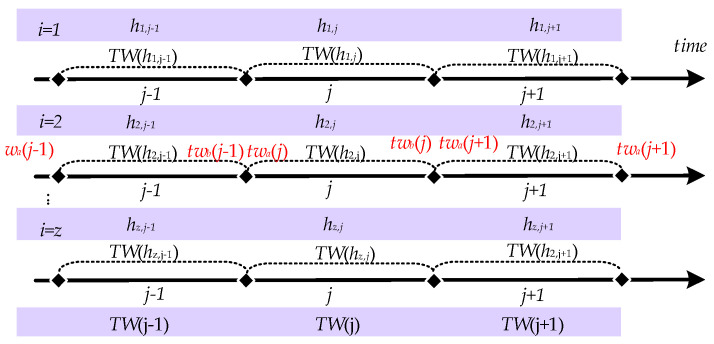
Description of Time Window.

**Figure 3 entropy-25-00116-f003:**
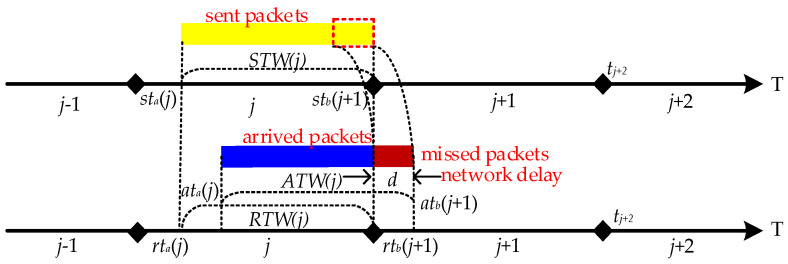
Delayed packet loss may be lost during network hopping.

**Figure 4 entropy-25-00116-f004:**
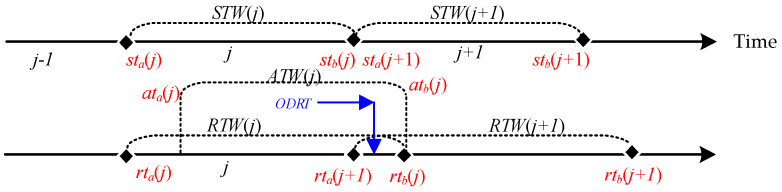
*ODRT* for overlapping data reception time.

**Figure 5 entropy-25-00116-f005:**
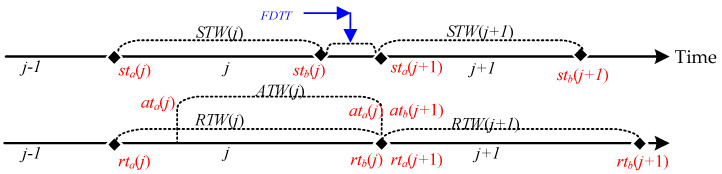
*FDTT* for idle data sending time.

**Figure 6 entropy-25-00116-f006:**
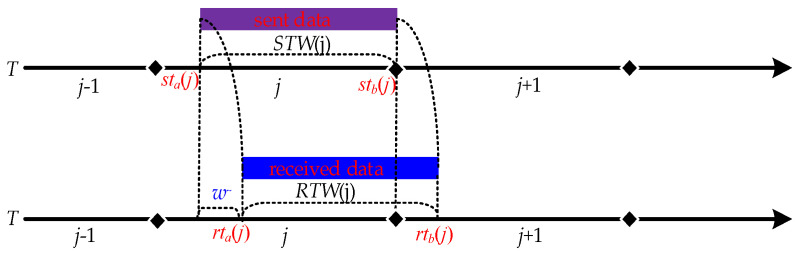
Diagram of the delay time window.

**Figure 7 entropy-25-00116-f007:**
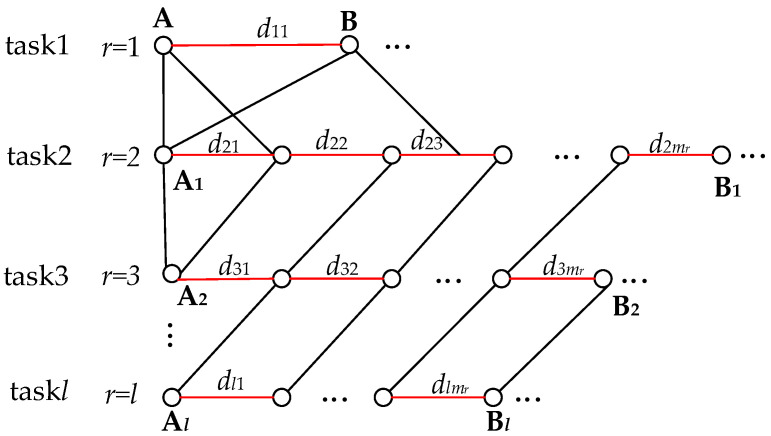
Whole connected network consisting of two or more nodes connected in a serial or parallel network.

**Figure 8 entropy-25-00116-f008:**
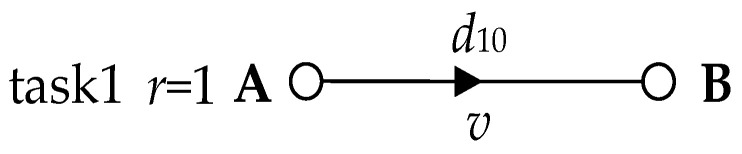
In the absence of intermediate nodes, end-to-end network transmission.

**Figure 9 entropy-25-00116-f009:**
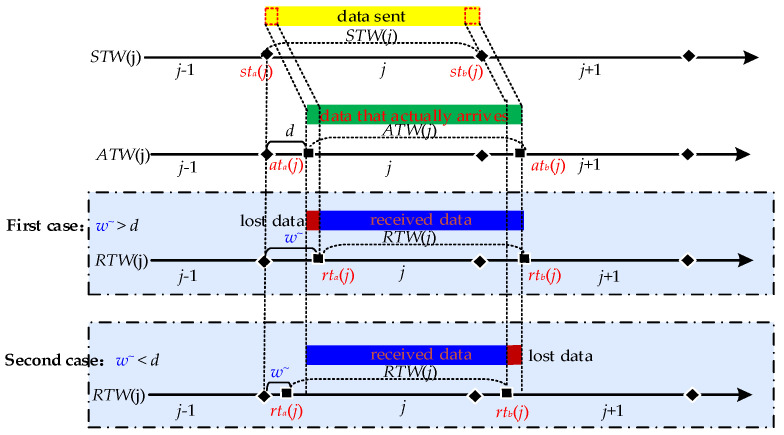
Data packet loss may occur when *w^~^ > d* or *w^~^ < d*.

**Figure 10 entropy-25-00116-f010:**
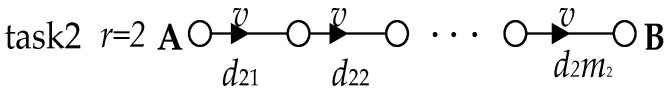
Multi-node network transmission.

**Figure 11 entropy-25-00116-f011:**
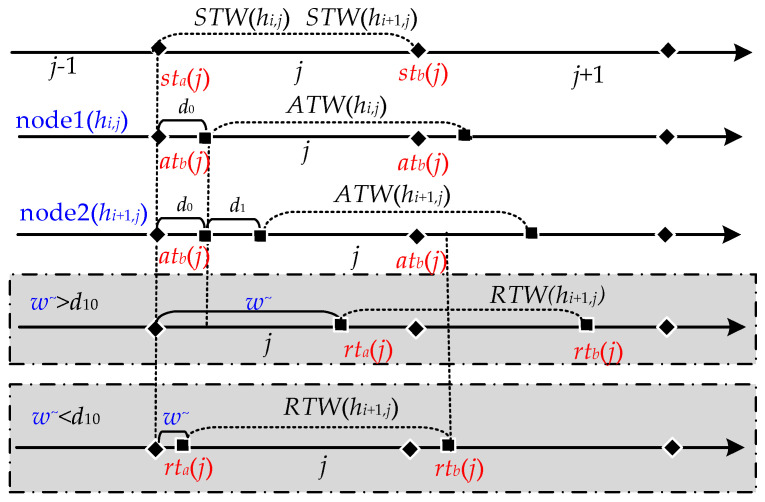
Delayed packet loss for two nodes based hopping network.

**Figure 12 entropy-25-00116-f012:**
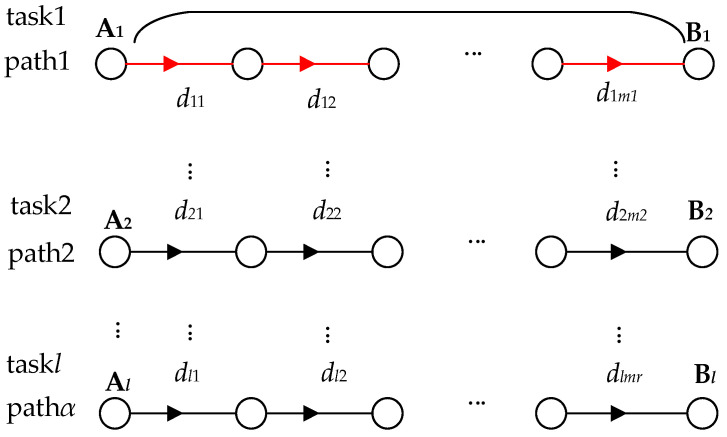
Network transmission for the entire network.

**Figure 13 entropy-25-00116-f013:**
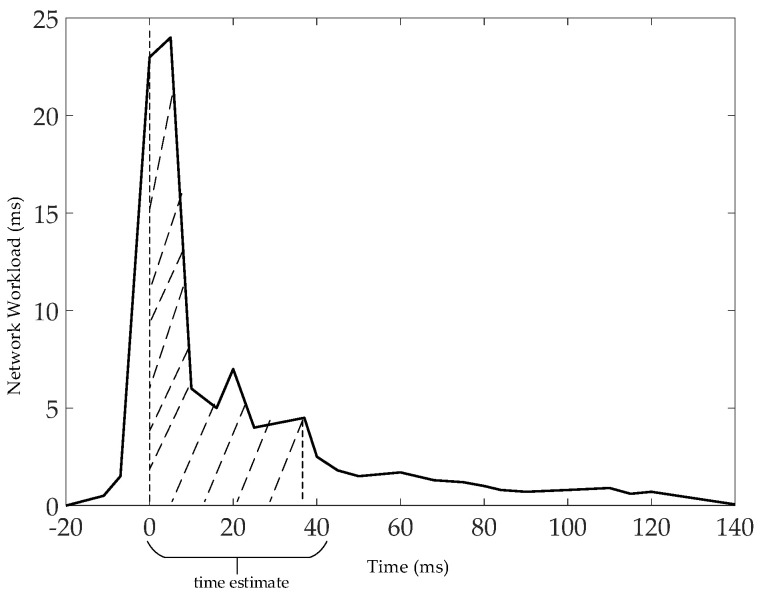
Distribution of $w_{n+1}-w_{n}-\delta$ at *δ* = 20 ms.

**Figure 14 entropy-25-00116-f014:**
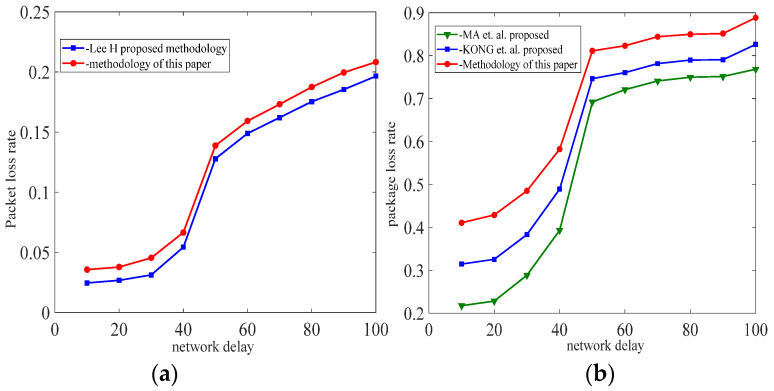
(**a**) Port hopping, (**b**) IP address hopping, the packet loss rates for end-to-end network. Refs [16,17].

**Figure 15 entropy-25-00116-f015:**
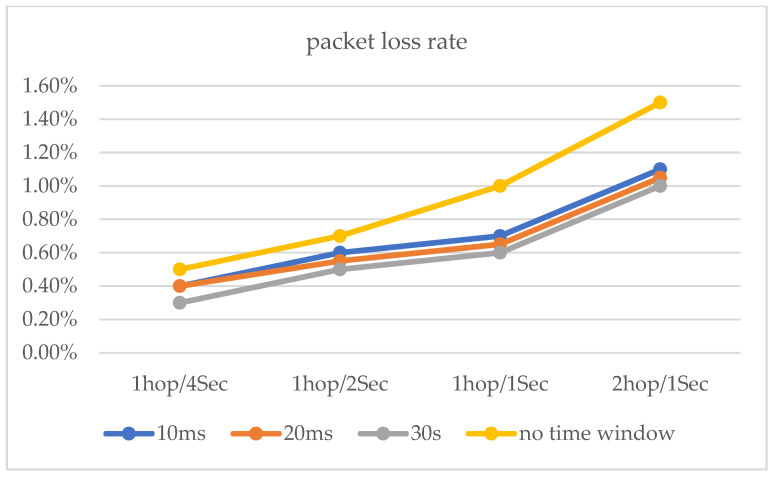
The packet loss rates with delay time window and no delay time window.

**Figure 16 entropy-25-00116-f016:**
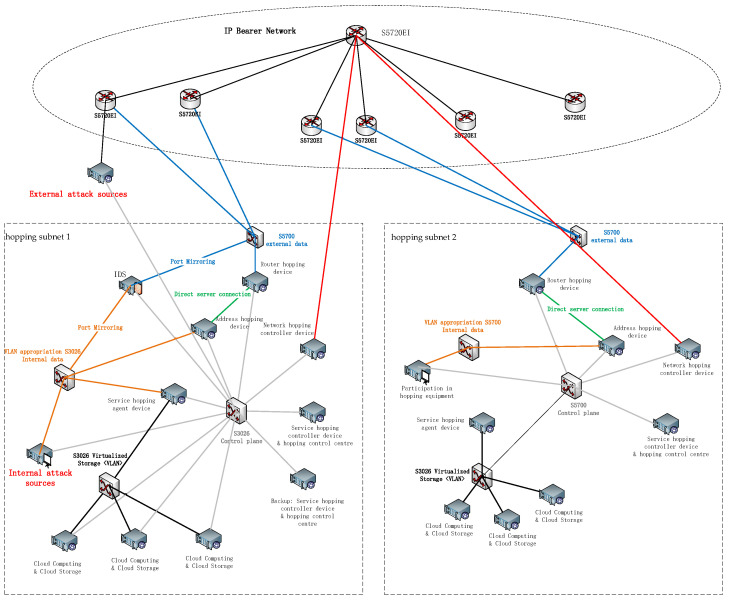
Physical connection topology of the hopping network.

**Figure 17 entropy-25-00116-f017:**
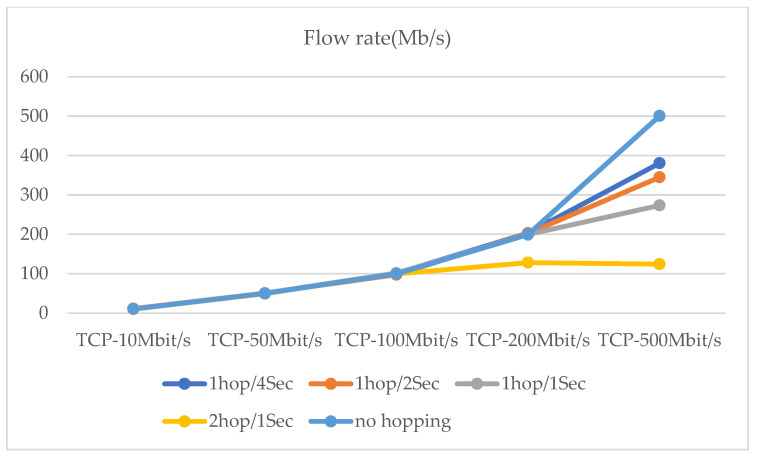
At 5 different hopping speeds, network transmission stress test.

**Figure 18 entropy-25-00116-f018:**
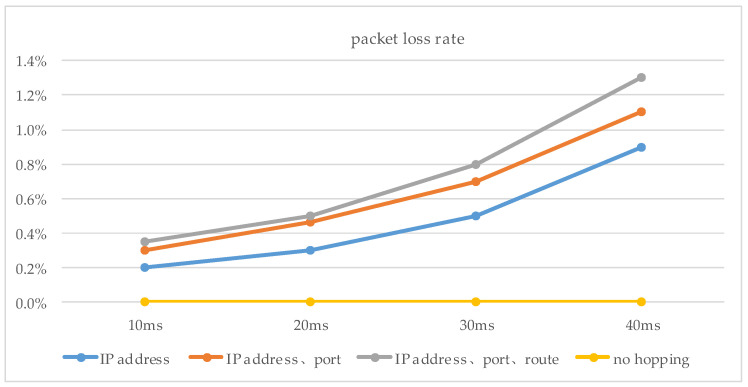
The packet loss rates with delay time window and no delay time window.

**Figure 19 entropy-25-00116-f019:**
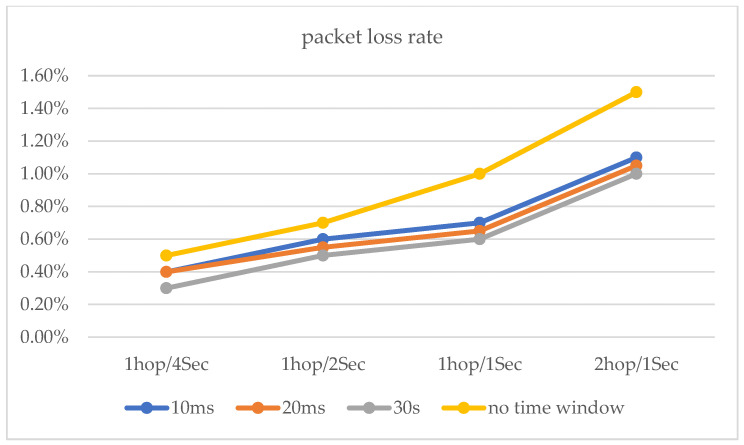
Transmission rate for 500 Mb/s, TCP data transfer.

**Table 1 entropy-25-00116-t001:** Comparison of packet loss rates for different time window methods.

Method	*m* = 1	*m* = 2	*m* = 3	*m* = 4	*m* = 5
Ours	0.3	0.35	0.5	0.6	0.8
*ODRT* [14]	0.6	0.7	0.8	0.9	1.0
*FDTT* [15]	0.8	0.9	1.1	1.2	1.4
*FDTT* [16]	0.9	1.2	1.4	1.6	1.7

**Table 2 entropy-25-00116-t002:** List of hardware devices and their configuration parameters for Cabinet 1, Cabinet 2 and Cabinet 3.

	Device Name	Device Type	Hardware Parameter	Network Interface
1	Cloud computing and cloud storage	server (Think-station)	CPU: E5-2609; memory: 16G; harddisk: 500G*5	2* Gigabit Ethernet port
2	Cloud computing and cloud storage	server (Think-station)	CPU: E5-2609; memory: 16G; harddisk: 500G*5	2* Gigabit Ethernet port
3	Cloud computing and cloud storage	server (Think-station)	CPU: E5-2609; memory: 16G; harddisk: 500G*5	2* Gigabit Ethernet port
4	Management platform	server (Think-station)	CPU: E5-2609; memory: 16G; harddisk: 500G*3	2* Gigabit Ethernet port
5	Network hopping controller	server (Think-station)	CPU: E5-2609; memory: 16G; harddisk: 500G*3	2* Gigabit Ethernet port
6	Service hopping controller	server (Think-station)	CPU: E5-2609; memory: 16G; harddisk: 500G*3	2* Gigabit Ethernet port
7	Service hopping agent	server (FitServer)	CPU: E5-2609; memory: 16G; harddisk: 500G*2	2* Gigabit Ethernet port
8	Address hopping devices	server (FitServer)	CPU: E5-2609; memory: 16G; harddisk: 500G*2	2* Gigabit Ethernet port
11	Control plane connection	Switche S3026	24-port 2-Layer switch	2* Gigabit Ethernet port
12	Control plane connection	Switche S5700 (Li)	24-port 2-Layer switch	2* Gigabit Ethernet port
13	Cloud computing and cloud storage connection	Switche S5700 (Li)	24-port 2-Layer switch	2* Gigabit Ethernet port

## Data Availability

Not applicable.

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
