# Peer review of "Network Delay and Cache Overflow: A Parameter Estimation Method for Time Window Based Hopping Network"

_entropy, 2023, doi:10.3390/e25010116_

Round 1
Reviewer 1 Report
Page 1.
Line 38 should be “defense”
Line 39 “firstly” should be “first”
Line 39 “that has been 39 presided ” should be “which was presided”
Line 40 “with the new” should be “ with new”
Page 2.
Line 46 Is “isomory” a real word?
Line 53 “one” should be “some”
Line 79 “Lee H” should be “Lee”
Line 82 “KONG” should be “Kong”
Line 83 “MA” should be “Ma”
Page 3
Line 118 “networ” should be “network”
Page 4
Line 142 “change, i.e.” should be “change”
Line 144 “completely i.e.” should be “completely”
Line 149 “one is” should be “One is”
Page 5
Line 165 Caption needs to be on same page as the figure.
Line 166 should be “Proposed”
Page 9
line 296 Replace “Jean-Chrysotome Bolot ” by “Bolot ”
Line 297 Replace “Under the assumption that the assumption of using”
by “Under the assumption of using ”
Line 314 The graph needs vertical axis label to remove bracket
Page 10
Line 316. The label needs to be on the same page as the graph.
Page 11
Line 345 “develope” should be “develop”
Page 15
Line 437 “zhu Fang ” should be “Zhu Fang”
Line 455 “TECHENOLOGIES” should be “TECHNOLOGIES”
Page 16. should be checked for consistency. Some names use initials. Some give full first names.
Q: What does the paper have to do with entropy?
Perhaps use of free version of Grammarly would be helpful.
Author Response
Thanks for your valuable comments on our manuscript. We have revised our manuscript according to your comments.

Reviewer 2 Report
In this study, the authors proposed a delay time window and a method for estimating the delay time window, and obtained network delay estimates for different data tasks and use them as estimates of the delay time window, and validate the estimated results to verify that the results satisfy the delay distribution law. I find that the idea is inspiring, the research is well designed and conducted, and the results are sound. The paper has the potential to be accepted for publication. Before that, the authors are advised to consider the following comments and suggestions. Therefore, I recommend a major revision for this submission.
1. A notation list to tabulate different symbols/abbreviation used is required.
2. In the paper, the figure title is too concise, please explain the figure and its symbol labeling in detail. And check the correctness of labeling in the figure, such as Chinese characters in Fig4 and 5, 30s in Fig15 and 18, and so on.
3. In part 3, what are the specific external factors. The authors show the external factors can be neglected and whether this is too idealistic leading to influence the correctness of the results.
4. In Section 4.2, is there simply only simulation results and no results comparing the theoretical framework with the simulation? If not, please specify. If so, how can the correctness of the previous theoretical model be verified?
5. In Part 4, the authors consider 10ms, 20ms, 30ms, and no time window for the proposed delay time window method to conduct experiments, in practical applications, what delay time window should be selected as the most appropriate?
6. It is suggested to modify the article format according to the requirements of the journal.
7. The English of this manuscript should be improved furthermore. It is better to invite a native speaker to edit the whole paper.
Author Response
We are very grateful to the editor's comments. We have revised it in accordance with the comments.

Round 2
Reviewer 2 Report
The authors addressed the reviewer's comments. It seems to improve a lot.
I recommend accepting.